# Food Taste, Dietary Consumption, and Food Preference Perception of Changes Following Bariatric Surgery in the Saudi Population: A Cross-Sectional Study

**DOI:** 10.3390/nu13103401

**Published:** 2021-09-27

**Authors:** Nora A. Althumiri, Mada H. Basyouni, Faisal Saeed Al-Qahtani, Mohammed Zamakhshary, Nasser F. BinDhim

**Affiliations:** 1Sharik Association for Health Research, Riyadh 13326, Saudi Arabia; mada.basyouni@sharikhealth.net (M.H.B.); Zamakhshary@alumni.utoronto.ca (M.Z.); nasser.bindhim@sharikhealth.net (N.F.B.); 2Ministry of Health, Riyadh 11176, Saudi Arabia; 3Department of Family & Community Medicine, College of Medicine, King Khalid University, Abha 13329, Saudi Arabia; drfaisalqahtani@gmail.com; 4College of Medicine, Alfaisal University, Riyadh 11533, Saudi Arabia; 5Saudi Food and Drug Authority, Riyadh 13513, Saudi Arabia

**Keywords:** food taste changes, food preference changes, bariatric surgery, obesity surgery, eating behavior, food choices

## Abstract

Background: Bariatric surgery is known as the most effective treatment resulting in long-term weight loss for obesity. However, behavioral changes, including food preference, food allergies, and consumption, between groups of patients who underwent bariatric surgery in comparison with people who did not have bariatric surgery have not been fully discussed in the literature. Objective: The aim of this article is to describe patient-reported changes of perception related to food preferences, consumption, and food allergies in participants who underwent bariatric surgery and to compare their food consumption with participants who did not have bariatric surgery in Saudi Arabia. Methodology: This study is a secondary analysis of the Sharik Diet and Health National Survey (SDHNS) conducted in July 2021. Quota sampling was utilized to generate balanced distributions of participants by age and gender across all administrative regions of Saudi Arabia. Data collection included sociodemographic information (age, gender, and educational level), as well as food habits and the consumption of various food categories. Results: Of the 6267 potential participants contacted in 2021 from the 13 administrative regions of Saudi Arabia, 5228 successfully completed the interview, with a response rate of 83.4%. Gender was distributed equally among the total participants in the sample. The prevalence of bariatric surgeries in Saudi Arabia was estimated at around 4.1% of the total sample. More than 36% of people who had bariatric surgery experienced food taste changes, and around 15% reported a decrease in allergic reactions to food. Moreover, 68.1% had food preference changes, either starting to favor a new food or no longer favoring one. There was a significant association between consuming more red meat, chicken, and energy drinks and a decreased consumption of grains and rice among those who had the bariatric surgery compared with those who did not. Conclusions: This study found that more than two-thirds of people who had bariatric surgery self-reported food taste and food preference changes. More studies should be performed on the Saudi population, including clinical follow-up, to better understand these changes.

## 1. Introduction

Obesity is a worldwide epidemic affecting children, adolescents, and adults [1]. Obesity prevalence tripled between 1975 and 2016, and its association with other diseases has become well known [2]. Recent studies in 2020 showed that the prevalence of obesity in Saudi Arabia ranged between 24.7% and 25.1% [3,4]. Bariatric surgery, such as Roux-en-Y gastric bypass, is known as the most effective treatment resulting in superior long-term weight loss compared with other measures for those with severe obesity, morbid obesity, or obesity with comorbidities who do not respond to other weight management programs [5]. Bariatric surgery is associated with the reduced incidence of new diabetes, hypertension, and hypercholesterolemia, in addition to ensuring prolonged weight loss and improving quality of life [6,7]. Globally, around 580,000 people undergo bariatric surgery annually [8]. The American Society for Metabolic and Bariatric Surgery (ASMBS) estimated that in 2019, more than 256,000 individuals in the United States underwent a bariatric surgery [6]. In Saudi Arabia alone, it is estimated that more than 20,000 bariatric surgeries are performed annually [9].

The results of these surgeries are due to several complex physiological and behavioral mechanisms that we are just beginning to understand, and more knowledge is needed in this domain [10]. The physiological mechanisms may include bile flow alteration, adipose hormone modulation, flow of nutrients, restriction of stomach size, as well as an altered anatomy, vagal manipulation, and enteric gut [11]. The behavioral changes may include taste, consumption patterns, and food preferences that appear or change after the bariatric surgery [10,12,13]. Although the restrictive nature of bariatric surgery would be expected to lead to a decrease in portion sizes, a change in preferences might suggest a change in underlying physiological responses to food [11,14].

Behavioral changes after bariatric surgery related to food preference, food taste, food allergies or intolerance, and food consumption have been noted in many published studies. For instance, the desire for sweet, high-carbohydrate, and fast foods after surgery significantly decreased when compared with normal weight controls indifferent to the amount of weight lost [15]. Changes in food taste, smell, or tolerance have also been reported after bariatric surgery [16]. One study found that around 97% of people experience at least one such change after these surgeries [17]. In terms of food allergies, bariatric surgery may change the digestion and hence allergenicity of food [18]. That might lead to systemic allergic reactions in people who, before bariatric surgery, experienced no or only mild oral allergy symptoms to these proteins [19]. A study in Saudi Arabia showed that there was a significant association between participants self-reporting food allergies and having bariatric surgery, in which participants in the bariatric surgery group reported fewer food allergies compared with the general population [20]. However, one study showed that there are no significant risk factors for developing new food allergies after such a surgery [21,22].

To our knowledge, no study has described the perception changes of food preference, taste, and consumption of people after bariatric surgeries compared to participants who did not have bariatric surgery in Saudi Arabia. Thus, the aim of this study was to describe patient-reported changes related to food preferences, consumption, and food allergies of people who underwent bariatric surgery and to compare their food consumption with participants who did not have bariatric surgery.

## 2. Methods

### 2.1. Study Design

This study is a secondary analysis of the Sharik Diet and Health National Survey (SDHNS) conducted in July 2021. The SDHNS is an annual nationwide cross-sectional survey conducted through phone interviews in Saudi Arabia [23].

### 2.2. Sampling and Sample Size

The SDHNS uses a proportional quota sampling technique to acquire an equal distribution of participants stratified by age and gender across the 13 regions of Saudi Arabia. Based on the median age of Saudi Arabian adults (36 years), two age groups (18–36 and 37+) were used, leading to a sample of 52 strata. The ZdataCloud^®^ data collection system, which has integrated eligibility and sampling modules, was used to control the sample distribution [24]. The SDHNS sample size was calculated based on a medium effect size of 0.25, and participants were selected to empower the sample to compare between quota groups, with an 80% power and a 95% confidence level [25]. Thus, each quota (age/gender/region quota) required 100 participants, and the grand targeted sample size was 5200 participants.

### 2.3. Participant Recruitment

SDHNS participant recruitment was limited to Arabic-speaking adults and Saudi residents who were ≥18 years old and above. The maximum age of participants was 90 years old, generated via a random phone number list from the Sharik Association for Research and Studies [26]. Participants were contacted by phone on up to three occasions. If a participant did not respond, the number of a new participant with similar demographics (age, gender, and region) was generated until the targeted quota was completed.

### 2.4. Variables and Outcome Measures

The SDHNS includes sociodemographic information (age, gender, and educational level), as well as food habits and consumption of various food categories.

The SDHNS asked participants if they had bariatric surgery. If yes, then participants were asked a set of questions including how long it has been since the participant had the surgery, whether the participant experienced any food taste or food preference changes after the bariatric surgery, where the participant started to like a new type of food that they did not like before or stopped preferring a type of food that they used to like before the surgery, and whether the participant was able to eat any food that used to give them a food allergy before the bariatric surgery. In addition, we asked all participants about their food consumption of each food type per week. For example, “In the last 7 days, how many days did you consume at least one portion of seafood?”. The possible answers were categorized in 8 categories (never, 1 day during the last week, 2 days during the last week, 3 days during the last week, 4 days during the last week, 5 days during the last week, 6 days during the last week, 7 days during the last week). In terms of food habits, we asked all participants about how many days in the last week they ate breakfast and how many days in the last week they ate food prepared outside the home.

### 2.5. Data Analysis

Descriptive statistics were used to summarize all the variables. Chi-square analyses were used to explore the differences between those who had surgery and those who did not. A *p* value of <0.05 was used to indicate statistical significance. Data management and analyses were carried out using the Statistical Package for Social Sciences (SPSS v21, Armonk, NY, USA).

## 3. Results

Of the 6267 potential participants contacted in 2021 from the 13 administrative regions of Saudi Arabia, 5228 successfully completed the interview (i.e., success rate of 83.4%). Gender was distributed equally among the total participants in the sample (50% male and 50% female). The mean age was 36.8 and ranged from 18 to 90 years. The mean body mass index for the participants who had bariatric surgery was 27.4 kg/m^2^, and was 26.0 kg/m^2^ for those who had not. Table 1 shows the demographic distribution.

### 3.1. Prevalence of Bariatric Surgeries and Changes in Food Preference and Allergies

The prevalence of bariatric surgeries in Saudi Arabia was estimated at around 4.1% of the total sample. More than 36% of people who had the surgery self-reported food taste changes. Around 15% of participants reported a decrease in allergic reactions to food after the surgery. Table 2 shows the prevalence of bariatric surgeries and changes in food preference and allergies in Saudi Arabia.

### 3.2. Association between Time since the Surgery and Changes in Food Preference and Allergies

Food taste change was significantly associated with time since surgery. Table 3 shows the effect of time since the surgery on food preference changes and food allergy reactions.

### 3.3. Food Consumption Comparison between Bariatric Surgery Participants and Those Who Did Not Have Bariatric Surgery

Food consumption between non-bariatric surgery participants and participants who had bariatric surgery is shown in Table 4. The sample mean consumption for red meat was 3.04, chicken was 4.82, grain and rice 5.92, fresh juice 2.62, and energy drinks 1.78. However, the results showed that people who had bariatric surgery tended to consume less grain and rice and more red meat, chicken, and energy drinks. There was a significant association between consuming more than the recommended level of red meat, chicken, and energy drinks among those who had bariatric surgery compared with participants who did not have bariatric surgery (Table 4).

## 4. Discussion

This study used a nationwide cross-sectional survey to describe the food preferences, food allergies, and consumption of people who underwent bariatric surgery and compared their food consumption with participants who did not have bariatric surgery. The results showed that, following bariatric surgery, participants self-reported food preference and taste changes.

Some people reported that their preference changed to either favor food that they did not like before or the opposite. This might be due to physiological changes in responses to food after the surgery [14]. These changes could be due to an underlying functional change, such as a decrease in meal portion size, which is suggested to alter the gut hormone profile to the anorexic state [14]. This finding is consistent with other studies [11,27,28]. A study by the University Hospitals of Leicester on 103 patients who underwent gastric bypass found that 73% of participants reported taste changes [13]. One study found that studies that use survey instruments and questionnaires usually find that people self-report changes in taste, and many of them claim a change in food preferences [24]. Other studies have used clinical tests such as oral sampling and found little to no change in people’s ability to perceive taste or their preferences [24]. The relation to food preference could be partly due to intrinsic changes within the gustatory and olfactory systems after the bariatric procedures [11]. Therefore, self-reported changes in taste and food preferences generated via patient-reported outcomes require more in-depth clinical investigation to be validated and explained [27,28].

According to the present study, time since surgery could play a role in people’s food taste changes. This might be because the types of foods that they are now eating regularly have different sensory characteristics, so that, after six months, the foods they did eat before could be perceived as “more different” and, because of that, they report “changes in taste”. A systematic review has shown evidence that some intrinsic changes within the gustatory and olfactory systems following bariatric procedures could lead to an increase in sensitivity to sweet and fatty taste stimuli and a decrease in preference to sweet-tasting stimuli, as well as an increase in smell acuity [11]. Such changes could contribute to changing food preferences and thus to changes in the perception of taste after the surgery.

However, the participants claim a change in food allergy incidents after the surgery, with the majority of such effects occurring either at the first month or six months after surgery. These results suggest that changes in food taste, consumption, and allergy happen mostly in the first six months after the surgery, as people may be more comfortable trying new foods after their recovery, which results in changing their food habits. However, such changes may fade with time, as suggested by other studies [11,29]. A systematic review including 61 articles showed evidence supporting the changes in perception and hedonic taste following bariatric procedures [11]. Another study found that claims of “favorite foods” changed toward more healthy choices after surgery, but this effect lessened as time since surgery increased [29]. However, many studies have suggested that people who have undergone bariatric surgery have different food preference patterns depending on their sensory perceptions, the duration of their follow-up, and the success of bariatric surgery [30].

Although few studies have examined food intake patterns after surgical weight loss, this study found that the participants who had bariatric surgery were consuming more foods that are rich in protein, such as red meat and chicken, compared with those who did not have surgery. This finding is consistent with other studies [11,31]. We also found a significant association between people who had bariatric surgery and a decrease in the consumption of grains and rice in Saudi Arabia. This finding is consistent with other studies that obtained similar results [31,32]. However, a study in 2014 conducted on 175 patients to evaluate life habits and diet quality (based on the specific food pyramid) of those who had undergone bariatric surgery and had been recovering for at least six months found a low consumption of protein, fruits, vegetables, and vegetable oils [32]. Moreover, the intake of carbohydrates and fats was higher than the recommended level established in the pyramid [32]. Such a contradiction in food consumption after the surgery may be due to sociocultural factors and may explain the food preference changes.

To our knowledge, this is the first study in Saudi Arabia to study the association between bariatric surgery and taste, food preference, and food allergy changes after the surgery, as well as to compare the food consumption of those who underwent the surgery with participants who did not have bariatric surgery. This study has various strengths and limitations. Its strengths include the national coverage of the sample and the balanced gender and age distribution. However, the study is limited by its cross-sectional design, which lacks temporality and the ability to determine the cause-and-effect order, such as the participants’ food consumption before the surgery. In addition, perception bias may occur in such self-reported methods of data collection for food taste and preferences. Finally, the findings of this study need to be validated via the replication of similar results using various research methods such as clinical and physiological studies and not only patient reported outcomes.

## 5. Conclusions

This study aimed to describe the self-reported perceived food taste change, food preference, food allergies, and consumption of people who had bariatric surgery compared with participants who did not have bariatric surgery in Saudi Arabia. The study found that food taste and preference perception had changes after the bariatric surgery in the majority of participants.

More studies should be performed on the Saudi population, including behavioral assessments such as on food frequency or food intake recall studies before and after surgery, in addition to clinical follow-ups, to better understand these changes.

## Figures and Tables

**Table 1 nutrients-13-03401-t001:** Participants’ characteristics (*n* = 5228).

Characteristics	*n* (%)
Age Groups	
36 years and below	2616 (50)
37 years and above	2612 (50)
Gender	
Male	2612 (50)
Female	2616 (50)
Education Level	
Less than a bachelor’s degree	2629 (50.3)
Bachelor’s degree and above	2599 (49.7)

**Table 2 nutrients-13-03401-t002:** Prevalence of bariatric surgeries and changes in food preference and allergies.

Characteristics	*n* (%)
Have you ever undergone bariatric surgery, such as stomach sleeve or gastric bypass, for the purpose of weight loss?	
Yes	213 (4.1)
No	5015 (95.9)
How long has it been since you had the bariatric surgery? (*n* = 213)	
One month	21 (9.9)
Between one and three months	22 (10.32)
Between three and six months	18 (8.45)
More than six months	152 (71.36)
Have you experienced any food taste changes after the bariatric surgery? (*n* = 213)	
Yes	78 (36.61)
No	135 (63.38)
Have you experienced any food preference changes after bariatric surgery where you started to like a new food type that you didn’t eat before?	
Yes	87 (40.84)
No	126 (59.15)
Have you experienced any food preference changes after bariatric surgery where you stopped eating a type of food that you used to eat?	
Yes	127 (59.6)
No	86 (40.4)
Have you experienced any food preference changes, either starting to favor a new food or no longer favoring one?	
Yes	145 (68.1)
No	68 (31.9)
Do you currently eat any food that used to cause you a food allergy before bariatric surgery?	
Yes	32 (15.02)
No	180 (84.50)

**Table 3 nutrients-13-03401-t003:** Association between time since the surgery and changes in food preference and allergies.

Characteristics	Less than One Month *n* (%)	Between 1 and 3 Months *n* (%)	Between 3 and 6 Months *n* (%)	More than 6 Months *n* (%)	(*p*-Value) Chi-Square
Any food taste changes after the bariatric surgery	10 (12.8)	11(14.1)	11(14.1)	46(59.0)	0.018
Any food preference changes after bariatric surgery where you started to like a new food type that you didn’t eat before	11 (12.6)	11(12.6)	7(8.0)	58 (66.7)	0.493
Any food preference changes after bariatric surgery where you stopped eating a type of food that you used to eat before	10 (7.9)	14 (11.0)	14 (11.0)	89 (70.1)	0.268
Any food preference changes, either starting to favor a new food or no longer favoring one	5 (6.6)	11 (14.5)	9 (11.8)	51 (67.1)	0.334
Eating any food that used to cause you a food allergy before bariatric surgery	11 (34.4)	7 (21.9)	3 (9.4)	11 (34.4)	<0.001

**Table 4 nutrients-13-03401-t004:** Food consumption comparison between general population and bariatric surgery participants.

Characteristics	Non-Bariatric Surgery Participants *n* (%)	Bariatric Surgery Participants *n* (%)	(*p*-Value) Chi-Square
Food Habits			
Eating breakfast every day	1653 (33.0)	60 (28.2)	0.144
At least one prepared meal outside home during the last week	3856 (76.9)	164 (77.0)	0.971
Food Group Consumption			
Dairy			
As recommended	998 (19.9)	34 (16.0)	0.157
Not as recommended	4017 (80.1)	179 (84.0)	
Seafood			
As recommended (Two portions/week)	746 (14.9)	35 (16.4)	0.533
Not as recommended	4269 (85.1)	178 (83.6)	
Red Meat			
As recommended (Up to three portions/week)	3091 (61.6)	146 (68.5)	0.042
Not as recommended	1924 (38.4)	67 (31.5)	
Chicken			
As recommended (Up to three portions/week)	2027 (40.4)	102 (47.9)	0.030
Not as recommended	2988 (59.6)	111 (52.1)	
Vegetables			
As recommended (Every day)	500 (10.0)	19 (8.9)	0.616
Not as recommended	4515 (90.0)	194 (91.1)	
Fruits			
As recommended (Every day)	394 (7.9)	13 (6.1)	0.349
Not as recommended	4620 (92.1)	200 (93.9)	
Grains and Rice			
As recommended (Every day)	1961 (39.1)	50 (23.5)	<0.001
Not as recommended	3054 (60.9)	163 (76.5)	
Fresh Juice			
At least one bottle/week	1826 (36.4)	59 (27.7)	0.010
Canned Juice			
At least one bottle/week	1849 (36.9)	75 (35.2)	0.623
Energy Drinks			
At least one bottle/week	1457 (29.1)	86 (40.4)	<0.001
Carbonated Drinks			
At least one bottle/week	3412 (68.1)	138 (64.8)	0.320

## Data Availability

Available from Sharik Association for Research and Studies upon request.

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
