# Peer review of "Food Taste, Dietary Consumption, and Food Preference Perception of Changes Following Bariatric Surgery in the Saudi Population: A Cross-Sectional Study"

_nutrients, 2021, doi:10.3390/nu13103401_

Round 1
Reviewer 1 Report
OVerall, this is a large cross-sectional study of a Saudi Arabian population on taste and dietary changes after bariatric surgery. Although the surgical comparison (4%) is low, the total volume of the survey and respondent rate is impressive. My only comment is it would be most helpful to know the dominant procedure performed (Sleeve, Bypass?) in the cohort, or procedure % distribution if you have this data and do a sub-analysis, similar to your time analysis, to determine if the nutrient and taste changes were different between procedure types.
Author Response
Reviewer #1:
1- Overall, this is a large cross-sectional study of a Saudi Arabian population on taste and dietary changes after bariatric surgery. Although the surgical comparison (4%) is low, the total volume of the survey and respondent rate is impressive. My only comment is it would be most helpful to know the dominant procedure performed (Sleeve, Bypass?) in the cohort, or procedure % distribution if you have this data and do a sub-analysis, similar to your time analysis, to determine if the nutrient and taste changes were different between procedure types.
Authors’ Response: Noted with many thanks for reviewing this article. Unfortunately, we did not collect this data. I totally agree with the reviewer on the importance of collecting this data in next year wave. Snice this project is annual national survey, we are planning to add it for the next year. We would be happy to provide a follow up letter to editor on this sub-analysis next year.
Thanks again for the valuable suggestion.
Reviewer 2 Report
The manuscript “Food Taste, Dietary Consumption, and Food Preference Changes Following the Bariatric Surgery in the Saudi Population: A Cross-Sectional Study” used a survey, in a representative sample of the population and based on the results try to conclude about changes in taste, intake, preferences and food allergy after bariatric surgery. Taking into consideration what the authors could evaluate, the title seems abusive, since the authors only can give information about what participants perceived to change. So, instead of “changes”, I suggest to consider “perception of changes”.
There is information lacking in methodology, namely about the questions to which participants answered to collect information about food habits, what leads to doubts about what the authors can really conclude or not.
The authors need to perform major corrections.
Abstract – “across the kingdom” - This expression is strange… I think most of the English needs to be revised by a native speaker
Abstract – conclusions “This study found that more than two thirds of post bariatric surgery patients food taste and preference have changed after bariatric surgery.” – example of sentence needing revision
Introduction “ranged between 21.7% and 25.1% in 2020” - please specify: range between..? These two percentages refer what each?
Introduction “The physiological mechanisms may include …” - I think this sentence would benefit from having references
Objectives “the behavioral changes” - I have doubts if this is the expression that better characterizes what was done… I suggest to replace by “changes related with food preferences and consumption and food allergies”.
Methods, point 2.2. - This includes 37 years old to what maximum age? 65 years old? or from 37 to any age? Because in participant recruitment it is only reported that participants had more or equal to 18 years old.
Methods point 2.4. - This information is very incomplete. Please specify the instruments used to assess the different parameters studied. How were food intake habits evaluated? In results section it appears that authors evaluated these last, but this is not obvious, when reading methodology, neither how this was. It would be good to have this part of the questionnaire as supplementary information, for example, to see the questions individuals’ answered. All these questions were open questions? More detail is needed.
Results - There are results about BMI? The authors present results for people who was submitted to bariatric surgery, but I think it would be important to know the BMI of individuals, in order to know how is the population that the authors name “general population”. This because authors are comparing bariatric surgery with potentially very different situations… We cannot say that bariatric surgery improves behavior, since it leads to some behaviors closer the “normal population” if we do not know if this “general population” is healthy…
Results – first paragraph – “with a response rate of 83.4%” - I think the authors mean a success rate of participation, right? As it is, we have doubts if the percentage refers to the participants that were contacted and that really participated by filling the questionnaires, or if this is the percentage of filling obtained for questionnaires. I suppose it is the first, but the English needs to be improved to clarify.
Results – point 3.3. - I think more description of the results that are presented in the table could be provided in the text. For example the consumption of grains and rice appears to be lower that the one recommended, in bariatric surgery patients… As well, more information about food habits needs to be provided…. What are the mean frequency of consumption of each “group” evaluated, etc.
Discussion – 3rd paragraph - please replace “taste changes” by “taste changes perceived”
Discussion- 3rd paragraph – allergies or intolerances - In this case, I see similar proportion of reduction both in the first period after surgery and after 6 months. So the authors need to re-formulate this part of the text.
Discussion 3rd paragraph - Another hypothesis is that people perceive changes in taste because they change food habits. Possibly the types of foods that they are now eating regularly have different sensory characteristics, so, after 6 months, the foods they did eat before could be perceived as “more different” and because of that they report “changes in taste”… With the type of data collected, this cannot be elucidated, so, I think that this needs to be considered
Discussion – 4th paragraph – “enhancement of consuming foods rich in protein, such as red meat and chicken, after the surgery” - I think these conclusions lack scientific validity… The authors are comparing bariatric surgery patients with a population that can be very variable. It would be good if authors could show more detail about the results obtained for the total population (the variation coefficient of responses for each food group) and more detail about bariatric surgery participants (the variation coefficient of responses from bariatric surgery patients)… The fact of having no information about how these individuals did eat before surgery also limits the conclusions about changes in behavior. They report that they feel changes, but they do not report what they eat before and what they eat now. The fact of having no information, in material and methods, about how this type of intake information was collected also difficult my analysis to what is being discussed.
End of discussion, when limitations are presented - The type of questions can also limit conclusions, because taste perception and food acceptance changes were based on what individuals perceived to change and this perception may be not totally right.
Conclusions – “food taste and preference changes” - I think this is abusive. The authors can conclude about what people perceived and not about really changes
Author Response
Reviewer #2:
- The manuscript “Food Taste, Dietary Consumption, and Food Preference Changes Following the Bariatric Surgery in the Saudi Population: A Cross-Sectional Study” used a survey, in a representative sample of the population and based on the results try to conclude about changes in taste, intake, preferences and food allergy after bariatric surgery. Taking into consideration what the authors could evaluate, the title seems abusive, since the authors only can give information about what participants perceived to change. So, instead of “changes”, I suggest considering “perception of changes”.:
Authors’ Response: Agree, with many thanks for review this work, the title has been updated as suggested.
- There is information lacking in methodology, namely about the questions to which participants answered to collect information about food habits, what leads to doubts about what the authors can really conclude or not.
Authors’ Response:. Agree, we updated the method section to clarify how data about food habits were collected.
- Abstract – “across the kingdom” - This expression is strange… I think most of the English needs to be revised by a native speaker
Authors’ Response:. Noted, the sentence was updated!
- Abstract – conclusions “This study found that more than two thirds of post bariatric surgery patients food taste and preference have changed after bariatric surgery.” – example of sentence needing revision
Authors’ Response:. Noted, this part has been revised and the final manuscript has been submitted to MDPI English service for proofreading.
- Introduction “ranged between 21.7% and 25.1% in 2020” - please specify: range between...? These two percentages refer what each?
Authors’ Response: This part was revised for more clarity.
- Introduction “The physiological mechanisms may include …” - I think this sentence would benefit from having references
Authors’ Response: Done, the reference was updated.
- Objectives “the behavioral changes” - I have doubts if this is the expression that better characterizes what was done… I suggest replacing by “changes related with food preferences and consumption and food allergies”
Authors’ Response: Done, the objectives was updated!
- Methods, point 2.2. - This includes 37 years old to what maximum age? 65 years old? or from 37 to any age? Because in participant recruitment it is only reported that participants had more or equal to 18 years old.
Authors’ Response: Done, the methodology section was updated. There were no upper limit for inclusion, however, the range of recruited participants were between 18 to 90 years old.
- Methods point 2.4. - This information is very incomplete. Please specify the instruments used to assess the different parameters studied. How were food intake habits evaluated? In results section it appears that authors evaluated these last, but this is not obvious, when reading methodology, neither how this was. It would be good to have this part of the questionnaire as supplementary information, for example, to see the questions individuals answered. All these questions were open questions? More detail is needed
Authors’ Response: The method section was updated to show the questions and the potential answers used in the survey.
- Results - There are results about BMI? The authors present results for people who was submitted to bariatric surgery, but I think it would be important to know the BMI of individuals, in order to know how is the population that the authors name “general population”. This because authors are comparing bariatric surgery with potentially very different situations… We cannot say that bariatric surgery improves behavior, since it leads to some behaviors closer the “normal population” if we do not know if this “general population” is healthy…
Authors’ Response: The average BMI for each group was included in the results. We agree with the reviewer that the use of “general population” might be misunderstood by the readers, thus, we updated it to participants who did not had to bariatric surgery.
- Results – first paragraph – “with a response rate of 83.4%” - I think the authors mean a success rate of participation, right? As it is, we have doubts if the percentage refers to the participants that were contacted and that really participated by filling the questionnaires, or if this is the percentage of filling obtained for questionnaires. I suppose it is the first, but the English needs to be improved to clarify.
Authors’ Response: Agree, the results section was updated. Yes it is the percentage refers to the participants that were contacted and that really participated by answering the questionnaires.
- Results – point 3.3. - I think more description of the results that are presented in the table could be provided in the text. For example, the consumption of grains and rice appears to be lower that the one recommended, in bariatric surgery patients
Authors’ Response: Done, the section 3.3 was updated!
- As well, more information about food habits needs to be provided…. What are the mean frequency of consumption of each “group” evaluated, etc.
Authors’ Response: Done, we included the means of the significant category in the text of section 3.3!
- Discussion – 3rd paragraph - please replace “taste changes” by “taste changes perceived”
Authors’ Response: Done, the sentence was updated!
- Discussion- 3rd paragraph – allergies or intolerances - In this case, I see similar proportion of reduction both in the first period after surgery and after 6 months. So, the authors need to re-formulate this part of the text.
Authors’ Response: Done, discussion was updated!
- Discussion 3rd paragraph - Another hypothesis is that people perceive changes in taste because they change food habits. Possibly the types of foods that they are now eating regularly have different sensory characteristics, so, after 6 months, the foods they did eat before could be perceived as “more different” and because of that they report “changes in taste”… With the type of data collected, this cannot be elucidated, so, I think that this needs to be considered
Authors’ Response: Agree. Thank you we included your suggestion in the discussion.
- Discussion – 4th paragraph – “enhancement of consuming foods rich in protein, such as red meat and chicken, after the surgery” - I think these conclusions lack scientific validity… The authors are comparing bariatric surgery patients with a population that can be very variable. It would be good if authors could show more detail about the results obtained for the total population (the variation coefficient of responses for each food group) and more detail about bariatric surgery participants (the variation coefficient of responses from bariatric surgery patients
Authors’ Response: Agree we corrected the sentence to reflect clearly was actually found.
- The fact of having no information about how these individuals did eat before surgery also limits the conclusions about changes in behavior. They report that they feel changes, but they do not report what they eat before and what they eat now.
Authors’ Response: Agree, limitation part was updated.
- The fact of having no information, in material and methods, about how this type of intake information was collected also difficult my analysis to what is being discussed.
Authors’ Response: Done, the methodology section was updated!
- End of discussion, when limitations are presented - The type of questions can also limit conclusions, because taste perception and food acceptance changes were based on what individuals perceived to change and this perception may be not totally right.
Authors’ Response: Done, the limitation section was updated!
- Conclusions – “food taste and preference changes” - I think this is abusive. The authors can conclude about what people perceived and not about really changes.
Authors’ Response: Agree, the conclusion section was updated!
Thanks again for the valuable comments and suggestions.
Round 2
Reviewer 2 Report
The authors were able to address most of the concerns. Even so, I have some minor points that I consider important to be corrected:
Introduction - In the part where authors are presenting percentages, despite the correction, I still think the way it was presented is not fully clear. Instead of presenting this way, I suggest: “…ranged between 24.7% [3] and 25.1% [4].
Methods - point 2.4. - I still think that adding the questionnaire as supplementary material would be important.
Results - 1st sentence -
I think is not totally clear yet. I suggest: “…5228 successfully completed the interview (i.e. success rate of 83.4%).”
Results - when BMI data is presented, please add the units.
Discussion - 3rd paragraph -
I acknowledge the authors had accepted the suggestion, but the sentence I wrote needs to be supported with bibliography showing that changes in types of foods consumed, after bariatric surgery, happens. As it is, this is only a suggestion without support fro the known literature. Please add examples of studies reinforcing this suggestion.
Author Response
Reviewer #2:
- The authors were able to address most of the concerns. Even so, I have some minor points that I consider important to be corrected:
Authors’ Response: Noted with many thanks.
- Introduction - In the part where authors are presenting percentages, despite the correction, I still think the way it was presented is not fully clear. Instead of presenting this way, I suggest: “…ranged between 24.7% [3] and 25.1% [4].
Authors’ Response: Noted and updated as suggested.
- Methods - point 2.4. - I still think that adding the questionnaire as supplementary material would be important.
Authors’ Response: noted a translated version of the questions used is now submitted as a supplementary material.
- Results - 1st sentence - I think is not totally clear yet. I suggest: “…5228 successfully completed the interview (i.e., success rate of 83.4%).”
Authors’ Response: Noted and updated as suggested.
- Results - when BMI data is presented, please add the units.
Authors’ Response: Noted and updated as suggested with many thanks.
- Discussion - 3rd paragraph - I acknowledge the authors had accepted the suggestion, but the sentence I wrote needs to be supported with bibliography showing that changes in types of foods consumed, after bariatric surgery, happens. As it is, this is only a suggestion without support from the known literature. Please add examples of studies reinforcing this suggestion.
Authors’ Response: Agree, more information were included to address this point.
Thanks again for the valuable comments and suggestions.